# Bioinformatic Assessment of Factors Affecting the Correlation between Protein Abundance and Elongation Efficiency in Prokaryotes

**DOI:** 10.3390/ijms231911996

**Published:** 2022-10-09

**Authors:** Aleksandra E. Korenskaia, Yury G. Matushkin, Sergey A. Lashin, Alexandra I. Klimenko

**Affiliations:** 1Kurchatov Genomics Center, Institute of Cytology and Genetics, Siberian Branch of the Russian Academy of Science, Lavrentiev Avenue 10, 630090 Novosibirsk, Russia; 2Institute of Cytology and Genetics, Siberian Branch of the Russian Academy of Science, Lavrentiev Avenue 10, 630090 Novosibirsk, Russia; 3Department of Natural Sciences, Novosibirsk National Research State University, Pirogova St. 1, 630090 Novosibirsk, Russia

**Keywords:** protein abundance prediction, translation elongation efficiency, translation in prokaryotes

## Abstract

Protein abundance is crucial for the majority of genetically regulated cell functions to act properly in prokaryotic organisms. Therefore, developing bioinformatic methods for assessing the efficiency of different stages of gene expression is of great importance for predicting the actual protein abundance. One of these steps is the evaluation of translation elongation efficiency based on mRNA sequence features, such as codon usage bias and mRNA secondary structure properties. In this study, we have evaluated correlation coefficients between experimentally measured protein abundance and predicted elongation efficiency characteristics for 26 prokaryotes, including non-model organisms, belonging to diverse taxonomic groups The algorithm for assessing elongation efficiency takes into account not only codon bias, but also number and energy of secondary structures in mRNA if those demonstrate an impact on predicted elongation efficiency of the ribosomal protein genes. The results show that, for a number of organisms, secondary structures are a better predictor of protein abundance than codon usage bias. The bioinformatic analysis has revealed several factors associated with the value of the correlation coefficient. The first factor is the elongation efficiency optimization type—the organisms whose genomes are optimized for codon usage only have significantly higher correlation coefficients. The second factor is taxonomical identity—bacteria that belong to the class Bacilli tend to have higher correlation coefficients among the analyzed set. The third is growth rate, which is shown to be higher for the organisms with higher correlation coefficients between protein abundance and predicted translation elongation efficiency. The obtained results can be useful for further improvement of methods for protein abundance prediction.

## 1. Introduction

It is well-known that proteins are the key elements that provide cell function, hence many physiological processes are controlled by the efficient allocation of the cellular proteome [1]. That is why quantification of protein abundance is of great importance for medical and biological studies. Experimental methods for measuring the amount of protein are expensive and labor-intensive; therefore, the problem of predicting the amount of protein based on genetic data is urgent.

### 1.1. Protein Abundance Prediction Tools

There are several approaches for prediction of protein abundance and tools that are used to calculate protein abundance for a particular organism based on mRNA abundance data and parameters of mRNA sequence. Many of them are species-specific, such as the tool that was developed for predicting protein abundance of *Saccharomyces cerevisiae* and *Schizosaccharomyces pombe* [2]. The calculations are based on experimentally obtained mRNA abundance, codon usage, the mRNA folding energy, and proteins’ half-life, which were obtained for both organisms and used as constants for each protein. The correlation between predicted and measured protein abundance is 0.77 and 0.74 for *S. cerevisiae* and *S. pombe*, respectively. The authors underline the factors that highly impact prediction accuracy: mRNA abundance, codon usage, and energy of mRNA fold. Another example is a tool that uses mRNA data for the prediction of protein abundance for immune cells of humans and mice [3]. The correlation between predicted and measured protein abundance is about 0.79–0.94.

Also, a few tools working without mRNA abundance data exist. There is an algorithm that uses only mRNA sequence features data to predict highly and lowly expressed proteins of *S. cerevisiae*. This algorithm uses 91 features including various estimates of codon bias calculated in various ways, CG content, codon-pair frequencies, etc. It shows a high correlation coefficient (0.75) between predicted and measured data for the organisms under study; however, the application of this model to *Escherichia coli* shows a modest (0.5) correlation between measured and predicted data [4]. These algorithms show high accuracy on organisms that have been used for training. However, predicting basal protein levels in a more general case for non-model organisms still remains challenging. Moreover, even for such a well-studied model organism as *Escherichia coli*, different studies emphasize different factors contributing to actual protein abundance. According to [5], the major part of protein abundance (53%) is determined by transcript level and at least 12% of protein abundance is determined by effectors of translation elongation. On the other hand, translational initiation determines about 1% of protein abundance. However, there is other evidence suggesting that translation initiation also might play a significant role, especially, when the expression levels of individual genes are under focus rather than global translation efficiency and cellular fitness [6]. Therefore, bioinformatic estimation of factors contributing to protein abundance, such as elongation efficiency, is an important step towards in silico prediction of protein abundance levels. Here we investigate the capability to predict protein abundance for phylogenetically diverse organisms by using the EloE (Elongation Efficiency) tool [7,8,9,10]. This tool calculates parameters that impact elongation efficiency of the translation stage: codon bias, number of secondary structures in mRNA, and energy of secondary structures in mRNA.

### 1.2. Codon Usage Bias Impacts Elongation Efficiency

The codon usage bias is the unequal proportion of synonymous codons occurrence throughout a genome. The fact that codon usage bias is associated with gene expression level was shown by Sharp, Tuohy, and Mosurski [11], who demonstrated on yeast that the highly expressed genes consist of common codons and lowly expressed genes contain infrequent codons. The experiment with replacement of common codons to infrequent codons into *E. coli* genes showed that infrequent codons could increase translation time [12]. Per-codon elongation rates are crucially dependent on the tRNA pool [13]. Codons with low abundance of the corresponding tRNA require more time for the correct accommodation of the corresponding tRNA in the A-site of the ribosome, which makes them non-optimal codons. Therefore, the content of optimal and non-optimal codons across mRNA affects gene elongation efficiency. The fact that codon bias is associated with the level of gene expression has been shown in many studies and reviews [14,15,16,17,18,19,20,21,22,23]. The great impact of codon bias on translation efficiency was shown on *E. coli* [16,20], *S. cerevisiae*, and *Trichomonas vaginalis* [14]. The codon usage bias has proven to be a good predictor of gene expression for *S. cerevisiae* [4] and *Trypanosoma brucei* [19]; however, it proves to be insufficient for a number of other organisms including *Rickettsia*, *Ehrlichia*, *Buchnera, Mycoplasma, Micrococcus, Helicobacter*, and some spirochaetes [24,25,26,27,28,29,30,31].

There is a classical approach to quantify codon usage bias known as the Codon Adaptation Index (CAI) [32]. CAI is widely used to assess the codon usage bias in various organisms and is implemented in several programming packages [33,34,35].

The codon usage bias might tune gene expression according to gene function. As it has been shown for several *S. cerevisiae* genes, mRNAs of housekeeping genes, such as those involved in glycolysis, are uniformly enriched in high-optimality codons, whereas proteins involved in transient responses to stimuli, such as the pheromone response, are enriched in non-optimal codons [36]. Genes that control circadian rhythms in various organisms, including cyanobacteria and *Neurospora crassa*, are also deficient in optimal codons. The authors suggest that enrichment in non-optimal codons helps to provide low levels of such regulatory gene products to allow quick elimination of the product after stimulus has disappeared [36]. Frumkin with colleagues [18] recoded genes in *E. coli* to demonstrate that codon usage patterns not only tune the elongation rates of particular genes, but also affect the global protein translation efficiency. Optimal codon composition of highly expressed genes increases the efficiency of translation and, consequently, reduces the number of ribosomes required for expression of these genes. This allows to indirectly increase the rate of translation initiation for other transcripts due to an increase in the pool of free ribosomes. Similar results have been shown on *S. cerevisiae* [37]. Shah et al. also demonstrated on *S. cerevisiae* that codon bias strongly affects protein abundance in genes with high mRNA level, whereas the effect of codon bias on protein abundance in genes with low mRNA abundance (<1% of transcriptome) is much lower but still significant [38]. The latter might be accounted for by the fact that translation of low-expressed genes slightly contributes into the pool of free ribosomes.

Besides the factors listed above, codon usage bias is associated with a number of other factors. It can be related to the gene function, e.g., hydrophobic loci of encoded proteins are associated with specific codon usage and signal peptides demonstrate non-optimal codons enrichment, or to the location of gene on the chromosome strand (leading or lagging) [22].

In conclusion, optimization by codon content is crucially important for the highly expressed genes, especially the genes expressed constitutively, which are supposed to be optimized by codon content, but it might be less critical for other genes, especially those with low expression and under special regulation. Therefore, it is proven that codon bias is a good predictor of efficiency of translation elongation and gene expression levels for several organisms, which provides opportunities for prediction of gene expression at the constitutive level based on the codon bias of genes.

### 1.3. Secondary Structures Impact Elongation Efficiency

It has been shown that the translation elongation rate is tuned not only by the codon usage bias but also by mRNA secondary structures [39,40,41]. Strong mRNA secondary structures reduce the speed of translating ribosomes [37,38,42], as it requires time for unweaving [43]. It is provided by the helicase activity of the ribosome using the active mechanical unwinding mechanism [43,44,45,46]. In this mechanism, the ribosome is translocated by applying force to the closed state of the mRNA duplex, which requires additional energy consumption. This mechanism, revealed in the data obtained from *E. coli*, affects the basal rate of translocation in a prokaryotic cell [43]. Moreover, it is known that different ORFs on an intra-operon level translate differentially varying in rates as much as 100-fold, which was demonstrated for *E. coli* [47]. Apparently, minimization of the abundance and energy of secondary structures in mRNA is supposed to increase the translation elongation rate.

However, it should be noted that the abundance and the energy of mRNA secondary structures tend to be higher over coding regions compared to untranslated regions, as was shown in yeast and *E. coli* [48]. Secondary structures in mRNA perform various functions [39,40], including modulation of folding for some proteins [49], regulation of mRNA half-life [36,50,51,52], regulation of translational frameshifting [53,54], termination–insulation and re-initiation control [55], whereby secondary structures can influence other stages of gene expression. This can introduce uncertainty about the effect of mRNA secondary structures on protein abundance. However, since for the abovementioned reasons, the number and stability of mRNA structures negatively affect translation efficiency, and the impact of secondary structures in the implementation of the listed functions may vary among different taxonomic groups (for example, there are many other mechanisms for regulation of mRNA decay in bacteria [52]), we can assume that the number and energy of secondary structures in mRNA can be a good predictor of the protein abundance for some organisms.

It is interesting that codon usage might be associated with the stability of secondary structures. It has been shown for *E. coli* and *S. cerevisiae* that the regions of high secondary structure content are preferentially enriched in high-optimal codons while non-optimal codons are located in low structured regions. Authors suggest that this pattern allows compensation for their independent effects on translation, helping to smooth overall translational speed and reducing the chance of potentially detrimental points of excessively slow or fast ribosome movement [56]. Moreover, genes tend to have significant codon bias in the regions of extremely high and low levels of secondary structure, which is found across all domains of life [57]. As has been shown in yeast, both codon usage bias and mRNA structural stability positively regulate mRNA expression levels and, moreover, highly structured and stable mRNA is selected [58]. It seems that codon bias and secondary structures in mRNA tend to be balanced to ensure optimal level of gene expression.

### 1.4. Summary

In conclusion, codon bias and secondary structures greatly impact translation elongation efficiency and contribute to gene expression. Therefore, a prediction of protein abundance based on these parameters seems to be a useful perspective. Here we analyze the capability to predict protein abundance using the EloE tool that calculates elongation efficiency indexes (*EEI*) based on these parameters. Previously, this tool was applied to show a significant correlation between *EEI* and gene expression for *S. cerevisiae* (0.79) and for *Helicobacter pylori* (0.28) [10]. As demonstrated for *H. pylori*, the correlation between gene expression and *EEI* increases with gene length, showing a maximum correlation (0.58) at a gene length of about 2200 bp [59]. In this work, we assess the correlation between *EEI* and gene expression at the protein level for various prokaryotes with diverse lifestyles, including archaea, obligate and opportunistic pathogenic bacteria, cyanobacteria, and species adapted to harsh environments (in particular, extremely acidophilic bacteria *Acidithiobacillus ferrooxidans*, and halophilic archaea *Halobacterium salinarum*).

## 2. Results

We have analyzed the correlation between protein abundance and base elongation efficiency index (*EEI*) value for various groups of microorganisms (see the details in the Materials and Methods section) and have investigated how this correlation depends on the following factors:Base *EEI* type, i.e., the mode of evolutionary optimization of translation exhibited by a particular genome;Taxonomical identity of an analyzed genome;Cell doubling time, i.e., microorganism’s reproduction rate;Mean (*M*) and standard deviation (R) of ranks of ribosomal protein genes measured on the base *EEI* scale.

Taking into account these factors allows us to study the structure of the sample, disentangling their impact on the correlation coefficient value between protein abundance and *EEI*.

Different genomic features in association with the obtained correlation coefficients (corr(PA|*EEI*)) between base *EEI* and protein abundance have also been analyzed. Neither genome length (r = −0.004, *p* = 0.85) nor number of genes (r = 0.01, *p* = 0.84) nor number of tRNAs (r = 0.36, *p* = 0.37) correlate significantly with the correlation coefficient between protein abundance and *EEI*. At the same time, such characteristics as number of ribosomal protein genes (r = 0.488, *p* = 1 × 10^−16^) and GC content (r = −0.394, *p* = 0.02) demonstrate significant correlation with the corr(PA|*EEI*). The number of ribosomal genes also correlates with the minimal doubling time of a microbe (Spearman’s correlation coefficient r = −0.428, *p* = 0.046).

To understand the representativeness of using proteomic data, we have calculated proteome coverage. Proteome coverage, which is a percentage of protein-coding genes presented in proteomic data, varies among samples. The median coverage per studying organism is 50.8 with the standard deviation 24.1. This means that for most of the analyzed organisms, the data used for analysis do not characterize the entire proteome, but do cover at least a significant part of it.

The correlation, coverage, minimal doubling time, *EEI* type, and mean (*M*) and standard deviation (R) values for each organism are demonstrated in Table 1.

Overall, the mean Spearman’s correlation coefficient between protein abundance and *EEI* calculated for the whole sample equals to 0.4 (the boxplot depicting corresponding descriptive statistics is shown in Figure 1). The majority of analyzed organisms, with the exception of *Neisseria meningitidis*, have shown a significant correlation between protein abundance and base *EEI* values. However, the correlation coefficient values vary greatly among the organisms.

This result means that predicting protein abundance solely based on elongation translation characteristics, such as those calculated by EloE, will have good accuracy for some organisms and poor accuracy for others. Further analysis aims to reveal the parameters that contribute to the correlation coefficients’ values.

### 2.1. Dependence of Correlation between Protein Abundance and the EEI from EEI Type

To determine the patterns of the correlation coefficients’ distribution among the organisms depending on their mode of evolutionary optimization of translation, we split the sample into several subsamples according to the genome’s base *EEI* type established by EloE (see Figure 2).

The highest correlation was obtained for the organisms belonging to the *EEI*1 type, which relies primarily on codon usage optimization for efficient translation. The correlation coefficients for organisms which were assigned to the *EEI*2 and the *EEI*4 types are significantly lower. The optimization of elongation efficiency types for these organisms were based on the optimization of number of secondary structures in mRNA for the *EEI*2 type, and codon usage and the optimization of number of secondary structures in mRNA for the *EEI*4 type.

It is important to note that the organisms belonging to the types other than the codon usage bias optimization only type (the *EEI*1 type) do not demonstrate higher correlation coefficients if elongation efficiency indices are calculated taking into account codon usage bias only, i.e., using the *EEI*1 formula (see Figure 3 and Table A1). The correlation coefficients between the *EEI*1 indices and protein abundance are significantly lower (*p* = 0.02, Welch’s *t*-test) than the correlation coefficients between the base *EEI* type and protein abundance for the organisms belonging to the type which minimizes the number of secondary structures (*EEI*2) (Figure 3a). They are also lower for *Pseudomonas aeruginosa*, which belongs to the type that considers only energy of secondary structures (*EEI*3, see Figure 3b), though we do not have enough sample size to deduce any extrapolations from here. Finally, the type that considers the codon usage bias and the number of secondary structures in mRNA (*EEI*4, Figure 3c) demonstrates higher corr(PA|*EEI*4) values than corr(PA|*EEI*1) at a trend level (*p* = 0.24). Thus, applying the approach that considers different elongation efficiency types allows improvement of the accuracy of predictions for those organisms that do not demonstrate a clear codon usage optimization pattern.

### 2.2. Dependence of Correlation between Protein Abundance and the EEI from Phylogeny

Phylogenetically distant organisms can have significant differences in the regulation of gene expression. Therefore, the significance of the effect of translation elongation factors on the overall level of gene expression may also differ among phylogenetically diverse organisms. In this regard, the ability to predict protein abundance based on the elongation translation characteristics can vary greatly for different phylogenic groups.

Below, we have mapped the analyzed strains onto a phylogenetic tree in order to reflect the diversity of phylogenetic groups represented in the analysis and to determine for which phylogenetic groups the prediction of protein abundance by EloE provides the most accurate results, which is demonstrated in Figure 4 rendered using iTol [60].

As one can see, the tree includes both species known for codon usage bias being a reliable measure of their translation elongation efficiency (such as *E. coli*), and those who have been shown to contravene that pattern (such as *H. pylori* and the representatives of *Mycoplasma* genus). Accordingly, the former belong to the *EEI*1 optimization type, while the latter are distributed to the other elongation efficiency optimization types, which take into account the effect of secondary structures in mRNA. Moreover, there are a number of new species that have not been studied in this regard before, and which, therefore, present a special interest.

The most represented taxa are phylum Firmicutes, namely, class Bacilli, and phylum Proteobacteria, in particular, class Gammaproteobacteria. Also, the mean correlation coefficients among the studied organisms of these taxa are 0.59 and 0.46, respectively, which are higher than the mean correlation coefficient for the entire dataset (0.4). Notably, most of the bacteria belonging to these classes belong to the *EEI*1 type, which show higher correlation coefficients. However, this difference is significant only for class Bacilli, compared with the other microorganisms from the analyzed set (Welch test, *p* = 2.2 × 10^−5^). Other taxa are represented by only a couple of species, if any, and their correlation coefficients corr(PA|*EEI*) are highly varied. The differences among correlation coefficients probably occur due to the different extents of influence of the codon usage bias and secondary structures on gene expression among species.

Thus, one can use elongation efficiency indices for a theoretical assessment of expected protein expression profile in the case of absence of proteomic data for a particular representative of one of those classes that demonstrate relatively high correlation between protein abundance and their base *EEI*, though biological implications of belonging to a specific elongation efficiency optimization type might vary depending on the particular taxa.

### 2.3. Dependence of Correlation between Protein Abundance and the EEI from Minimal Doubling Time

Doubling time as a characteristic reflecting reproduction rate varies greatly, both among various bacterial species and inside the same species if it grows in different conditions [61]. It is known that bacterial growth rates are correlated with ribosome abundance [62], and therefore it correlates with the entire translation rate due to reduction in active ribosome fraction during slow growth [63]. However, translation elongation maintains a significant rate even in poor nutrient conditions with slow bacterial growth [63], which enables cells to produce proteins crucial for surviving in harsh environments in a timely manner.

The prediction of protein abundance using elongation efficiency indices assumes that coding sequences of highly expressed genes, such as ribosomal protein genes, are heavily optimized compared to the genes with low level of basal expression. This means that if elongation efficiency is more evenly optimized because it is a less essential step in determining protein abundance than, for instance, a gene regulation, such an organism can demonstrate a reduced quality of protein abundance prediction. Higgs and Ran [64] found a low correlation between tRNA gene abundance and codon usage for most bacteria with high doubling time. They supposed that, although the translation is the limiting factor of division in fast-growing organisms, this is not the case for slow-multiplying organisms. Although their results could also be explained by the high impact of mRNA secondary structures in translation, this aspect is still worth being tested.

Also, it was demonstrated [65] that a prokaryotic growth rate is highly correlated with the codon usage bias. In fast-growing organisms, codon usage bias is more pronounced due to codon usage optimization, which is crucial since the tRNA pool becomes limiting at very high growth rates. Based on the codon usage bias of ribosomal protein, Weissman, Hou, and Fuhrman have predicted [66] the minimum doubling time for about 200,000 prokaryotes. Such an estimation of the growth rate divides prokaryotes into two groups, which fits their ecological roles. The first one is copiotrophs, consisting of fast-growing microbes that grow in nutrient-rich environments. The other is oligotrophs, represented by microbes that are adapted to low levels of nutrients and tend to have slow growth rates. Based on these results, authors have defined oligotroph as an organism for which a selection for rapid maximal growth is weak enough so that translation efficiency is not optimized by selection for optimized codon usage.

In the light of the listed above, a hypothesis can be formulated that protein abundance predictions will be less efficient for prokaryotes with the high minimal doubling time.

Indeed, one can notice (Figure 5a) an increase in the corr(PA|*EEI*) with a decrease in the minimum doubling time (DT), although bacteria with fast growth and a low correlation coefficient also exist. The Pearson correlation coefficient between corr(PA|*EEI*) and minimal doubling time for 25 organisms is r = −0.446 (*p* = 0.025). No relationship was found between the base *EEI* type and the doubling time. However, it is worth noting that slowly growing bacteria (with the DT ≥ 5 h) are mostly represented by *EEI* types which consider secondary structures (only one out of seven organisms belong to the *EEI*1 type). Consistent with previous studies, codon usage bias slightly reflects the gene expression profile for those six organisms, which is demonstrated by calculation of corr(PA|*EEI*) for each of the five *EEI* types (see Table A1). Considering the secondary structures enables us to reach higher (but still quite low) correlation coefficients.

We hypothesize that some prokaryotic species living in harsh environments could demonstrate a similar level of translation efficiency optimization throughout the genome. Such organisms are supposed to show a high minimum doubling time and lower translation elongation efficiency for ribosomal genes than fast-growing species. As mean elongation efficiency of ribosomal proteins is reflected by *M* values, we have compared them for fast-growing and slow-growing prokaryotes (Figure 5b).

The Welch test between *M* values of fast-growing organisms (with the minimal doubling time no more than two hours) and slow-growing organisms (with the minimal doubling time higher than five hours) has shown a significant difference (*p*-value = 7.059 × 10^−6^). The comparison of medium-growing organisms (with the minimal doubling time between two and five hours) and slow-growing organisms also has shown a significant difference for *M* values (*p* = 0.0002).

Notably, the lower correlation between protein abundance and elongation efficiency for organisms with higher minimum doubling time cannot be explained only by a weaker optimization of ribosomal protein genes in favor of other genes. If we do not consider elongation efficiency of ribosomal protein genes during the selection of the base *EEI* type by selection of the *EEI* type that shows higher correlation coefficients between protein abundance and elongation efficiency, which simulates the usage of the optimal group of highly optimized genes, the correlation coefficients do not necessarily rise. In particular, changing *EEI* type greatly increases (from 0.12 to 0.34 for *Acidithiobacillus ferrooxidans*, and from 0.36 to 0.46 for *Leptospira interrogans*) the correlation coefficient only for two of seven slow-growing organisms under study (see Table A1). In summary, the prediction of protein abundance is less efficient for slow-growing organisms, which can be explained by less pronounced differences in elongation efficiency optimization throughout the genomes of these organisms. In other words, translation elongation efficiency does not appear to be a limiting factor in determining protein abundance for slow-growing microorganisms.

### 2.4. Dependence of Correlation between Protein Abundance and the EEI from Elongation Efficiency of Ribosomal Protein Genes

As mentioned earlier, the ranks of ribosomal gene proteins, which contribute to the *M* (mean) and R (standard deviation) parameters, are used to determine a genome’s base elongation efficiency index type, which describes the mode of evolutionary optimization of translation in a particular genome in the most accurate way. Here we have examined how the correlation coefficient between the *EEI* and protein abundance depends on the *M* and R values for the base *EEI* type (see Figure 6).

The Pearson correlation coefficient between *M* and corr(PA|*EEI*) for 25 organisms is 0.7344 (*p* = 2.9 × 10^−5^). The Pearson correlation coefficient between R and corr(PA|*EEI*) is −0.454 (*p* = 0.022). This reassuring result indicates that the strategy of maximizing *M* and minimizing R that is used to determine the base *EEI* type in the EloE is the right way, which not only has a theoretical basis but also is substantiated by experimental data.

As these parameters are highly correlated with corr(PA|*EEI*), they could be used for estimating prediction potential (correlation coefficient between the *EEI* and protein abundance) for an organism under study. Also, these parameters are calculated by the algorithm itself and do not require the involvement of additional data, which makes them convenient enough to assess the efficiency of the algorithm.

For this purpose, a linear regression model has been built. The independent variable is represented by the *M* parameter only, since *M* and R parameters are highly correlated.
(1)Corr.coef=0.0432+0.0054∗M

The determination coefficient (R^2^) equals 0.35, and the mean squared error (MSE) equals 0.011. The test for significance of regression shows F > F-critical (10.36 > 4.2793), *p* = 0.038, which means that the regression model is statistically significant. In summary, the statistics shows that the model has a prediction power.

Using this formula with caution, and taking into account the observed range of *M* values, one could predict the expected correlation coefficient for another organism, which does not have enough data covering its protein expression profile.

In summary, we can use the EloE for a rough prediction of gene expression at the protein level. Taking into account the *EEI* type, doubling time, taxonomic identity, as well as the *M* and R parameters, allows us to derive an approximate estimate of the expected correlation coefficient between base *EEI* values and actual protein abundance.

## 3. Discussion

The gene expression is a multi-level process including various regulation on a transcriptional and translational level. The protein abundance reflects the overall effect of all the factors contributing to the gene expression, whereas each of these factors has its own particular share in this cumulative effect. One of the intriguing questions within this context is the problem of predicting the basal gene expression based on only partial information available, in particular, the genomic sequence data. This study focuses on investigating correlation between the translation elongation characteristics and proteomic data. As our analysis indicates, the mean correlation coefficient between protein abundance and base elongation efficiency index (*EEI*) calculated for the whole sample is not high, which was expected, since we are trying to predict the protein abundance based on the elongation efficiency, while the protein yield is also influenced by other stages, including the stage of transcription, translation initiation [6,15,67], and other factors such as half-life values of the respective protein and mRNA [15,50,52], as well as the protein’s structure and its resistance to proteases [68,69,70]. To the best of our knowledge, this is the first time such an analysis of correlation between protein abundance and different elongation efficiency measures has been performed based on the proteomics data for the prokaryotes belonging to such a range of taxonomic groups including non-model organisms and the organisms which are known for codon usage being an ineffective measure of translation elongation efficiency of their genes.

However, the correlation coefficients between protein abundance and the *EEI* values vary greatly among the organisms. The bioinformatic assessment of the factors affecting the correlation between protein abundance and elongation efficiency in prokaryotes has shown that there are several factors associated with the value of the correlation coefficient. The first is the *EEI* type—organisms that correspond to the *EEI*1 type, which takes into account codon bias only, have significantly higher correlation coefficients. Such a difference between these types could be explained by ambiguous [71] contributions of secondary structures to protein abundance. Although secondary structures in mRNA decrease ribosome velocity, they can protect mRNA from ribonucleases and, therefore, increase mRNA abundance. As a result, protein abundance could both decrease and increase under the influence of secondary structures. Thus, we should expect a lower prediction accuracy for organisms belonging to optimization types, for which secondary structures play a significant role in determining the protein abundance (*EEI*2, *EEI*3, *EEI*4, and, probably, *EEI*5). Unfortunately, among the organisms with available protein profiles, *Neisseria meningitidis*, the only one belonging to the *EEI*5 base type, do not show a significant correlation between protein abundance and *EEI* values—not only base ones but any *EEI* values, including classic codon usage bias. Therefore, we refrain from making any decisive conclusions about that particular optimization type. It is worth noting, however, that for those organisms under study, which fall into one of the optimization types (*EEI*2, *EEI*3, and *EEI*4) characterized by the role of mRNA secondary structures, applying their base elongation efficiency index allows us to reach higher correlation coefficients than if using *EEI*1, which represents classic codon usage bias. We believe that this indicates the complex nature and the role of translation elongation efficiency in determining protein abundance in these classes of organisms.

The second factor is taxonomic identity of an organism under study—such a class as Bacilli is among those characterized by the highest correlation coefficient between *EEI* and protein abundance. Using this information to derive estimates of expected correlation coefficients for the organisms that lack proteomic profiles seems to be a promising approach, though we definitely need more data to be able to improve the quality of such an assessment. The third factor is the microorganism’s reproduction rate. We observe an increase in the correlation coefficient between the *EEI* and protein abundance with a decrease in the minimum doubling time, that is, fast-growing prokaryotes tend to have a high correlation coefficient. The latter might be associated with the similar level of elongation efficiency across the genome in slow-growing species, which is reflected in ribosomal protein coding genes being not the most highly optimized group of genes among them. The fact that genes encoding ribosomal proteins may not be highly efficient at translation elongation was shown on several *Mycoplasma* species (*C. M. haemolamae*, *M. haemocanis*, *M. wenyonii*, *M. haemofelis*, *M. pneumonia*, *C. M. haemominutum*, and *M. suis*). These species demonstrate decreased M values and a reduced number of perfect local inverted repeats (potential hairpins) in mRNA of both ribosomal and non-ribosomal genes. It makes translation elongation efficiency of non-ribosomal genes similar to ribosomal ones [72]. Thus, there are various situations where either an organism possesses a quite compact and evenly optimized genome or translation elongation efficiency does not appear to be a limiting stage in determining protein level. However, we have also demonstrated that, in general, the initial approach used by the EloE that relies on assessing the ranks of ribosomal proteins in the gene list sorted by the base *EEI* values is adequate to the experimental data of the organisms under study, especially for the organisms with a high number of ribosomal protein genes and low GC content. Therefore, it can be used in further development of the algorithms that would take into account not only translation elongation, but also other stages that affect the level of gene expression.

One of the difficulties in studying the relationships between elongation efficiency characteristics and protein abundance at the organism level is the lack of the genome-wide protein abundance profiles to assess the actual correlation between protein abundance and elongation efficiency indices based on representative datasets, which would include protein-encoding genes with various expression levels for taxonomically divergent organisms, including non-model ones. However, as more proteomic studies generating a full protein profile of an organism under study are published, the whole picture of how the particular aspects of optimization of translation elongation efficiency affect the protein abundance in various microorganisms will become more clear and detailed. We believe that a thorough bioinformatic estimation of factors contributing to protein abundance, such as elongation efficiency, paying attention to the actual biodiversity of prokaryotic species, is an important step towards in silico prediction of protein abundance levels.

## 4. Materials and Methods

### 4.1. Proteomic and Genomic Data

Gene expression at the protein level data was taken from the PaxDb database [73]. This database stores proteomic data obtained by MS-MS spectroscopy, which provides quantitative protein abundance information. Proteomes were obtained from 26 prokaryotic organisms, including 24 bacteria and 2 archaea. Since for many organisms, numerous experimental data are present, in the further calculation we used a median abundance of each protein per organism. The genomes of these strains with the corresponding loci identifiers were obtained from the NCBI Assembly database [74], the list of species presented in Table 2.

*Neisseria meningitidis* has been excluded from the subsequent analysis due to its insignificant correlation between protein abundance and base *EEI* values.

### 4.2. EloE Elongation Efficiency Indices (EEI)

In this article, we have analyzed elongation efficiency indices calculated by EloE (Elongation Efficiency) tool [7] (developed by Sokolov V.S, Novosibirsk, Russia). The executable file of the tool as well as the user’s manual, input and output data are deposited in the Appendix A. It requires a Java Runtime Environment (JRE) SE 7 or higher to run the program. The EloE algorithm calculates elongation efficiency indices (*EEI*) for each organism’s protein-coding gene in five ways (Table 3) [9,10] using annotated genome sequence. The elongation efficiency indices are calculated according to Formula (2):(2)EEI(i)=K/(w1Ta(i)+w2Te(i))
where *K*—normalization constant, which is used to assure the range of *EEI*(*i*) within [0, 10]; *w*1 and *w*2—weight coefficients (equals 0 if a parameter is excluded and 1 if it is considered); Ta(i) estimates the average time required for the fixation of isoacceptor aminoacyl-tRNA in the A site of the ribosome; and Te(i) estimates the average time demanded by the ribosome for the translocation stage. There are two options for Te(i) function: $LCI_{L}\left(i\right)$, which calculates local complementarity based on number of potential secondary structures in mRNA, and $LCI_{E}\left(i\right)$, which calculates local complementarity taking into account the energy of potential secondary structures in mRNA. The formulae can be found in Appendix B.

For each elongation efficiency index (*EEI*), protein-coding genes are sorted in descending order according to the corresponding *EEI* values. To determine the index type that properly describes the efficiency of elongation translation in the particular organism under study, mean (*M*) and standard deviation (R) of the ranks of ribosomal protein genes are calculated. The ribosomal protein genes are known to be intensely expressed along a wide range of organisms and assumed to be, therefore, optimized in the efficiency of translation elongation. However, a list of highly expressed genes can be manually set by a user.

*M* and R values are calculated for each of the five *EEI* types [10].
(3)$$M_{rank}=\frac{1}{N_{rib}}\sum_{i=1}^{N_{rib}}x_{i};$$
(4)$$R_{rank}=\sqrt{\frac{1}{N_{rib}}\sum_{i=1}^{N_{rib}}{\left(M_{rank}-x_{i}\right)}^{2}};$$
$\mathrm{where}\;M_{rank}—$the mean rank of ribosomal protein genes, $R_{rank}—$the standard deviation of ribosomal protein genes’ ranks, $N_{rib}—$the number of ribosomal protein-coding genes, and $x_{i}—$is the rank of ribosomal protein-coding gene in the gene set arranged in order of increasing *EEI* values.
(5)$$M=100*\left(\frac{2*(M_{rank}-1)}{N_{tot}-1}-1\right),$$
(6)$$R=100*\frac{2*(R_{rank}-1)}{N_{tot}-1},$$
$N_{tot}$ is the total number of protein-coding genes, *M* is the normalized mean rank of ribosomal genes, and R is the normalized standard deviation for ranks of ribosomal genes calculated for the *EEI* type.

We regard the type that has the maximum *M* parameter as the base organism type. If there are several types sharing the maximum *M* value, a type with the minimum R is defined as the base type. Elongation efficiency indices for protein-coding genes of an organism are calculated by this type. In this study, we have analyzed *EEI*s of the base type for each organism from Table 2.

### 4.3. Statistical Analysis and Regression Model

To estimate the power of elongation efficiency indices as predictors of gene expression at the protein level, we calculated the correlation coefficient between experimentally measured protein abundance and base *EEI* for each organism; we will further refer to it as (corr(PA|*EEI*). As *EEI* indices have a rank-size distribution, we used Spearman’s rank correlation coefficient [75] with the p-value threshold for statistical significance of 0.05. Then we have also calculated the correlation between corr(PA|*EEI*) and other parameters (doubling time, *M*, R parameters). In this case, we have used Pearson’s correlation. Since the values do not correspond to the normal distribution, verification of statistical significance has been provided by the bootstrap method [76] using the “boot” package in R.

The linear regression model has been built for 25 samples (see Table 2). Predictor variable is represented by M parameter, the dependent variable is corr(PA|*EEI*). The linear regression model has been built using the Sklearn package in Python using the entire dataset. The quality of the model was assessed by R^2^, mean square error (MSE), and mean absolute error (MAE) using Monte Carlo cross validation from the cross_validate and ShuffleSplit functions from the Sklearn package with splitting the dataset 2000 times into 20% test and 80% training sets.

### 4.4. Minimal Doubling Time

The value of minimal doubling time for each organism has been obtained from the literature (see Table 4). If the doubling time differs depending on medium and temperature, the minimum value is selected. All values are turned into hours.

## Figures and Tables

**Figure 1 ijms-23-11996-f001:**
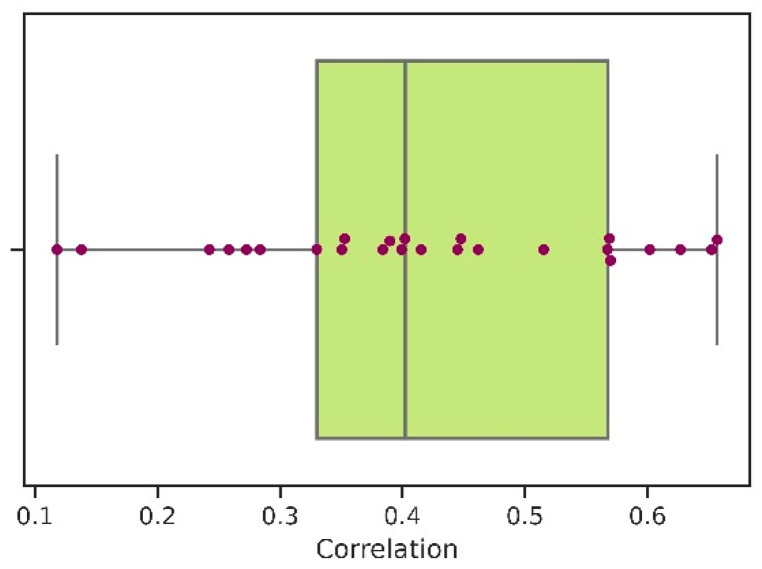
Spearman’s correlation coefficients between protein abundance and *EEI* for 25 prokaryotes.

**Figure 2 ijms-23-11996-f002:**
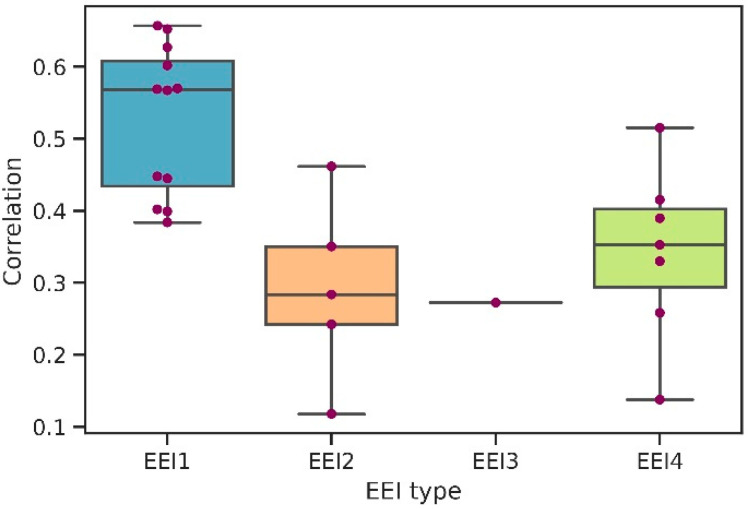
Spearman’s correlation coefficients between protein abundance and *EEI* distribution among 5 *EEI* types for 25 prokaryotes (*Neisseria meningitidis*, the organism with *p*-value > 0.05, is excluded).

**Figure 3 ijms-23-11996-f003:**
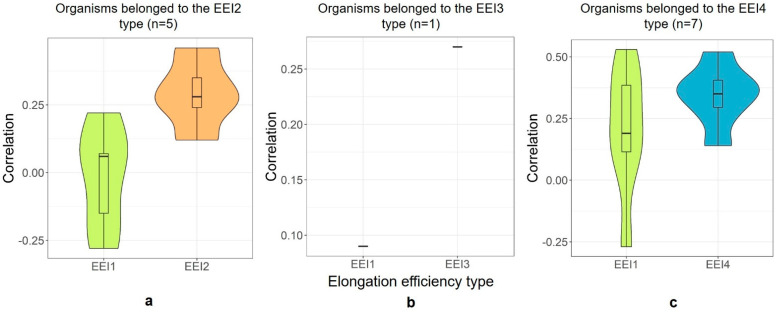
The distributions of the Spearman’s correlation coefficient values between protein abundance and *EEI* for the organisms belonged to the different elongation efficiency optimization types that take into account: number of secondary structures (*EEI*2, panel (**a**)), energy of secondary structures (*EEI*3, panel (**b**)), codon bias and number of secondary structures (*EEI*4, panel (**c**)). All these distributions are compared with the correlation between protein abundance and indices for codon bias-based index (*EEI*1) calculated for the same organisms.

**Figure 4 ijms-23-11996-f004:**
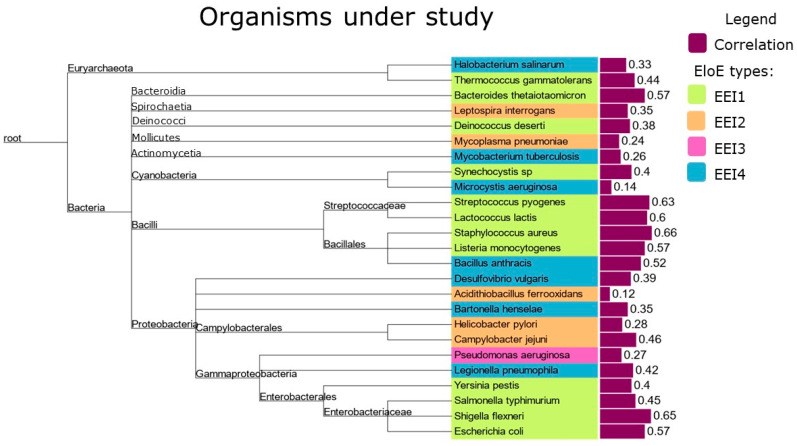
The distribution of the analyzed 25 organisms by taxonomical categories. Arrows point to corresponding higher-order taxa of the analyzed species. A number near a species name corresponds to the base *EEI* type and a colored square shows a Spearman’s correlation (corr(PA|*EEI*)) coefficient value (see the legend).

**Figure 5 ijms-23-11996-f005:**
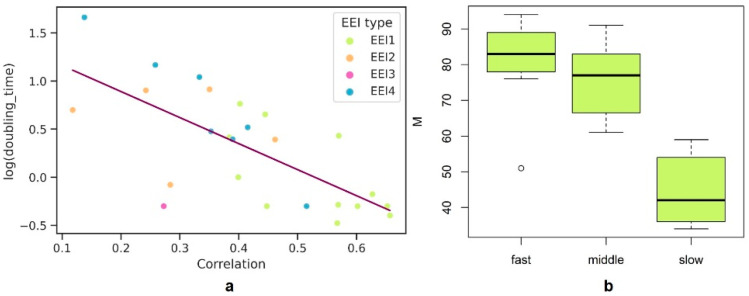
(**a**) Dependence of corr(PA|*EEI*) coefficient from the logarithm of minimal doubling time for 25 organisms. The color describes *EEI* type (see the legend). The trend line is colored purple; (**b**) distribution of bacteria with diverse growth rates (with the minimal doubling time <2 h, ≥2 h and <5 h, and ≥5 h for the fast, medium, and slow growing bacteria, respectively) by the M parameter.

**Figure 6 ijms-23-11996-f006:**
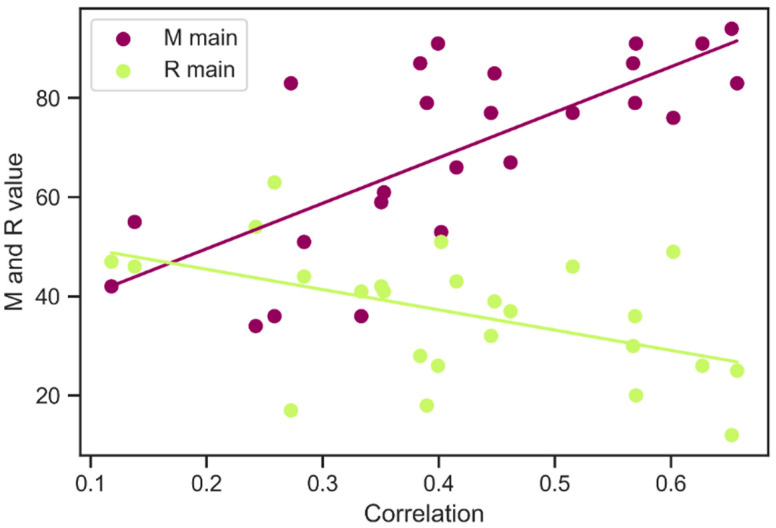
Dependence between corr(PA|*EEI*) and M (mean ribosome protein-coding gene rank) and R (standard deviation of ribosome protein-coding genes’ rank) parameters.

**Table 1 ijms-23-11996-t001:** Values of the analyzed parameters for the studied organisms: elongation efficiency type (*EEI* type), which was obtained by EloE (see the details in Materials and Methods section); coverage of proteomic data; Spearman correlation coefficients between protein abundance and base *EEI* index; corresponding *p*-value; minimal doubling time (see the references in Table 4); mean (M) and standard deviation (R) values of ranks of ribosomal protein genes measured on the base *EEI* scale.

Organism	*EEI* Type	Coverage	Correlation Coefficient	*p*-Value	Doubling Time (h)	M_Main	R_Main
*Staphylococcus aureus*	1	62.5	0.66	8.46 × 10^−211^	0.4	83	25
*Shigella flexneri*	1	39.4	0.65	4.01 × 10^−202^	0.5	94	12
*Streptococcus pyogenes*	1	75.9	0.63	3.60 × 10^−141^	0.6667	91	26
*Lactococcus lactis*	1	57.1	0.60	2.58 × 10^−128^	0.5	76	49
*Bacteroides thetaiotaomicron*	1	15.9	0.57	2.52 × 10^−67^	2.7	91	20
*Listeria monocytogenes*	1	16.4	0.57	1.46 × 10^−41^	0.5167	79	36
*Escherichia coli*	1	97.40	0.57	0	0.3333	87	30
*Bacillus anthracis*	4	26.20	0.52	3.42 × 10^−102^	0.5	77	46
*Campylobacter jejuni*	2	47.40	0.46	6.48 × 10^−42^	2.4667	67	37
*Salmonella typhimurium*	1	56.3	0.45	1.79 × 10^−126^	0.5	85	39
*Thermococcus gammatolerans*	1	62.2	0.44	3.71 × 10^−66^	4.5	77	32
*Legionella pneumophila*	4	25.2	0.42	2.50 × 10^−05^	3.3	66	43
*Synechocystis sp.*	1	37.8	0.40	8.01 × 10^−48^	5.8	53	51
*Yersinia pestis*	1	29.6	0.40	4.86 × 10^−47^	1	91	26
*Desulfovibrio vulgaris*	4	27.1	0.39	3.87 × 10^−37^	2.48	79	18
*Deinococcus deserti*	1	38.5	0.38	6.58 × 10^−48^	2.6	87	28
*Bartonella henselae*	4	85.7	0.35	1.21 × 10^−41^	3	61	41
*Leptospira interrogans*	2	66.2	0.35	6.06 × 10^−66^	8.2	59	42
*Halobacterium salinarum*	4	54.2	0.33	2.00 × 10^−02^	11	36	41
*Helicobacter pylori*	2	98.8	0.28	1.22 × 10^−29^	0.8333	51	44
*Pseudomonas aeruginosa*	3	43.6	0.27	1.37 × 10^−42^	0.5	83	17
*Mycobacterium tuberculosis*	4	84	0.26	3.44 × 10^−28^	14.7	36	63
*Microcystis aeruginosa*	4	79.00	0.24	1.60 × 10^−06^	46	55	46
*Mycoplasma pneumoniae*	2	60.9	0.14	1.14 × 10^−83^	8	34	54
*Acidithiobacillus ferrooxidans*	2	41.9	0.12	1.73 × 10^−05^	5	42	47

**Table 2 ijms-23-11996-t002:** The list of species under study (species for which proteomic data were collected) and corresponding assembly accessions.

№	Species	Assembly Accession
1	*Acidithiobacillus ferrooxidans ATCC23270*	GCF_000021485.1
2	*Bacillus anthracis str. Sterne*	GCF_000008165.1
3	*Bacteroides thetaiotaomicron VPI-5482*	GCF_000011065.1
4	*Bartonella henselae str. Houston-1*	GCF_000046705.1
5	*Campylobacter jejuni NCTC11168*	GCF_000009085.1
6	*Deinococcus deserti VCD115*	GCF_000020685.1
7	*Desulfovibrio vulgaris str. Hildenborough*	GCF_000195755.1
8	*Escherichia coli K12 MG1655*	GCF_000005845.2
9	*Halobacterium salinarum NRC-1*	GCF_000006805.1
10	*Helicobacter pylori 26695*	GCF_000008525.1
11	*Lactococcus lactis subsp. lactis Il1403*	GCF_000006865.1
12	*Legionella pneumophila subsp. pneumophila str. Philadelphia 1*	GCF_000008485.1
13	*Leptospira interrogans serovar Lai str. 56601*	GCF_000007685.1
14	*Listeria monocytogenes EGD-e*	GCF_000196035.1
15	*Microcystis aeruginosa NIES-843*	GCF_000010625.1
16	*Mycobacterium tuberculosis H37Rv*	GCF_000195955.2
17	*Mycoplasma pneumoniae FH*	GCF_001272835.1
18	*Neisseria meningitidis MC58*	GCF_000008805.1
19	*Pseudomonas aeruginosa PAO1*	GCF_000006765.1
20	*Salmonella enterica subsp. enterica serovar Typhimurium str. LT2*	GCF_000006945.2
21	*Shigella flexneri 2a str. 301*	GCF_000006925.2
22	*Staphylococcus aureus*	GCF_000009665.1
23	*Streptococcus pyogenes*	GCF_000006785.2
24	*Synechocystis sp. PCC 6803*	GCF_000009725.1
25	*Thermococcus gammatolerans EJ3*	GCF_000022365.1
26	*Yersinia pestis CO92 (enterobacteria)*	GCF_000009065.1

**Table 3 ijms-23-11996-t003:** The description of elongation efficiency types (*EEI* types) calculation.

Type	Codon Usage (Ta(i))	Local Complementarity Level (Potential mRNA Secondary Structures, $Te\left(i\right)\;Depending\;on\;LCI_{L}\left(i\right))$	Local Complementarity Level with the Energy of Potential mRNA Secondary Structures $\left(Te\left(i\right)\;Depending\;on\;LCI_{E}\left(i\right)\right)$
*EEI*1	+	—	—
*EEI*2	—	+	—
*EEI*3	—	—	+
*EEI*4	+	+	—
*EEI*5	+	—	+

**Table 4 ijms-23-11996-t004:** The table reflects minimal doubling time for each species in hours and in a logarithmic form. The article from which the data were taken is also presented for each of the organisms (column DT_source).

Organism	Doubling_Time (DT), h	Log (DT)	DT_Source
*Acidithiobacillus ferrooxidans*	5	0.69897	[77]
*Bacillus anthracis*	0.5	−0.30103	[78]
*Bacteroides thetaiotaomicron*	2.7	0.431364	[79]
*Bartonella henselae*	3	0.477121	[80]
*Campylobacter jejuni*	2.466667	0.39211	[81]
*Deinococcus deserti*	2.6	0.414973	[82]
*Desulfovibrio vulgaris*	2.48	0.394452	[83]
*Escherichia coli*	0.333333	−0.47712	[61]
*Halobacterium salinarum*	11	1.041393	[84]
*Helicobacter pylori*	0.833333	−0.07918	[85]
*Lactococcus lactis*	0.5	−0.30103	[86]
*Legionella pneumophila*	3.3	0.518514	[87]
*Leptospira interrogans*	8.2	0.913814	[88]
*Listeria monocytogenes*	0.516667	−0.28679	[89]
*Microcystis aeruginosa*	46	1.662758	[90]
*Mycobacterium tuberculosis*	14.7	1.167317	[91]
*Mycoplasma pneumoniae*	8	0.90309	[92]
*Pseudomonas aeruginosa*	0.5	−0.30103	[93]
*Salmonella typhimurium*	0.5	−0.30103	[94]
*Shigella flexneri*	0.5	−0.30103	[95]
*Staphylococcus aureus*	0.4	−0.39794	[93]
*Streptococcus pyogenes*	0.666667	−0.17609	[96]
*Synechocystis sp.*	5.8	0.763428	[97]
*Thermococcus gammatolerans*	4.5	0.653213	[98]
*Yersinia pestis*	1	0	[78]

## Data Availability

All data generated during this study are included in this published article.

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
