# Peer review of "Bioinformatic Assessment of Factors Affecting the Correlation between Protein Abundance and Elongation Efficiency in Prokaryotes"

_ijms, 2022, doi:10.3390/ijms231911996_

Round 1

Reviewer 1 Report

The manuscript by Korenskaia et al. describes a result of bioinformatics analysis in evaluating factors affecting the correlation coefficients between protein abundance and translation elongation efficiency in 26 prokaryotes. They claimed that they identified three factors associated with the value of the correlation coefficient, 1) the elongation efficiency optimization type, 2) taxonomical identity, and 3) the growth rate of the organisms. Although the factors would not fully explain the discordance between the protein abundance and the elongation efficiency for each protein-coding gene, these results would be helpful for understanding the complete picture of the determinant factors in protein abundance in prokaryotes.

1.     Are any genomic features such as genome size, the number of protein-coding genes, etc. associated with the value of the correlation coefficient?

2.     Do you have any idea about the reason why Neisseria meningitidis exceptionally do not show a significant correlation between protein abundance and base EEI values?

Minor:

The colors for EEI1 and EEI4 in Figures 3 and 4(a) would be better to be consistent.

Author Response

Thank you for your comments! We addressed the raised points in the revised version of the manuscript.

R1: Are any genomic features such as genome size, the number of protein-coding genes, etc. associated with the value of the correlation coefficient?

Response: We have additionally tested such features as the genome length, the number of genes in the genome, the number of tRNAs, the number of rRNAs, GC content and found that only a number of ribosomal protein genes and GC content correlate significantly with the value of the correlation coefficient between EEI and protein abundance.

R1: Do you have any idea about the reason why Neisseria meningitidis exceptionally do not show a significant correlation between protein abundance and base EEI values?

Response: It is an interesting question, however, it is difficult to determine the exact reason why Neisseria meningitidis do not show a significant correlation between protein abundance and EEI values (not only base ones but any EEI values, including classic codon usage bias). We are inclined to attribute this fact to the coverage of available protein abundance profile, which is quite low - the proteomic dataset covers only about 20% of protein-coding genes of Neisseria meningitidis – and it can be non-representative. We could answer this question definitely only if we got a protein abundance profile of higher coverage, which includes genes of various expressivity.

R1: The colors for EEI1 and EEI4 in Figures 3 and 4(a) would be better to be consistent.

Response: We agree and introduce respective adjustments into the manuscript.

Reviewer 2 Report

In this manuscript, the authors apply an old analysis program of their own to analyze the protein abundance on various organisms in relation to codon usage. It is loaded with an impressive number of numbers, but also has two main problems: (1) The novel contribution or practical usefulness of the study is not convincing; (2) The analyses are not tied to the biology or phylogeny of the organisms or the function of the genes. So, for example, what is the point of presenting the phylogeny in Fig. 3? Without any reference to the organisms, this Figure (and most of the paper) are just numbers. There is a verbal description under Figure 3 but it is just a summary, not any biological insight of the organisms in terms of EEI. “The most represented taxa” (line 249) is simply because the authors presented more of these taxa than the others. But did it skew the pattern? They also present very correlation between EEI and phylogeny. Inexplicably, however, the next paragraph states: “Thus, in future it could be assumed that since the studied bacteria belonging to this class have a high correlation between EEI and protein abundance, then other representatives of this class that are not presented in the analysis will also have a high correlation, and therefore a relatively high accuracy in predicting the amount of protein” !!

 The field of translational control by codon usage, tRNA abundance, mRNA structures and stability etc. is nearly saturated with both experimental data and bioinformatic analysis. Protein  abundance, additionally, is controlled by stability, dictated by the protein’s higher order structure   and resistance to proteases, a difficult and largely neglected area. Over several decades, a large number of highly capable investigators have proposed and tested every imaginable model of translational control, covering an exhaustive list of potential factors and their combinations that can influence the translational rate in many organisms. For every model, multiple exceptions and interpretations have also been noted. While this is in part due to multifactorial complexity of the translation regulation, it also requires that every new manuscript in this area must clearly and convincingly explain its need and originality. In that respect, I suggest that in the Introduction the authors add more focus on what is really new here. What is in it that no one has done or seen before. It will not be easy, but that is exactly why it will be helpful to the general readership.

 Ref 47 is specifically about intra-operon ORF. A global reference or references will be more appropriate here (Line 139-141). As regards Ref 48, the data are data, but the compensation interpretation (Line 150) has remained controversial.

 IJMS is an English journal. I doubt that Ошибка! Источник ссылки не найден in the legend of Table 1 will be helpful to its readers. My own effort to translate it to English in Google Translate gave me the following: “Oshibka! Istochnik ssylki ne nayden. Error! Reference source not found”. Authors: Please remember that you are writing the paper for the readers.

 The different “types” of EEI were never very clear, but this is a major point of the paper. EEI refers to the authors’ own application, which they named “Elongation Efficiency (Index) or EEI”. They cite it as published in 2015 (Journal of Integrative Bioinformatics), and that paper describes five types of EEI (1 to 5), instead of four (in the current manuscript). However, the authors have also stated, “The EloE web  application  is  available  at  http://www-bionet.sscc.ru:7780/EloE”. But this link does not open. The 2015 paper itself is also very cryptically written without much details of the either the application or the biological analyses. The 2015 paper in PDF cites yet another paper as a reference for EEI: V. A. Likhoshvai, Yu. G. Matushkin. Nucleotide composition-based prediction of gene expression efficacy. Mol. Biol. (Moscow), 34(3):345–350, 2000. But this paper cannot be found in PubMed. So, the road reaches a dead-end.

 The Discussion section is a list of factors that can cause uncertainties in their studies and very correctly so, but it ends with the following futuristic conclusion: “We believe that bioinformatic estimation of factors contributing into protein abundance, such as elongation efficiency, is an important step towards in silico prediction of protein abundance levels.” But this was already stated in their previous papers (as mentioned above) where the application was first described. The current manuscript was expected to extend it to actual biological relevance, which is again left for the future. Thus, the paper makes little or no advancement in this field.

 Ref 18 is rather old (2016). It should be replaced with more recent ones. Some examples are offered below, but the authors may know others:

 Review J Mol Evol . 2021 Dec;89(9-10):589-593. doi: 10.1007/s00239-021-10027-z. Epub 2021 Aug 12. Codon Usage Bias: An Endless Tale.

Andrés Iriarte, Guillermo Lamolle, Héctor Musto

PMID: 34383106 DOI: 10.1007/s00239-021-10027-z

Mol Biol Rep. 2022; 49(1): 539–565.

Published online 2021 Nov 25. doi: 10.1007/s11033-021-06749-4

PMCID: PMC8613526

PMID: 34822069

Codon usage bias

Sujatha Thankeswaran Parvathy, Varatharajalu Udayasuriyan, and Vijaipal Bhadana

Analysis of computational codon usage models and their association with translationally slow codons.

Gabriel Wright ,Anabel Rodriguez,Jun Li,Patricia L. Clark,Tijana Milenković,Scott J. Emrich

Published: April 30, 2020

https://doi.org/10.1371/journal.pone.0232003

Author Response

Thank you for a thorough review of our work! We revised the manuscript according to your comments and would like to address the raised points.

Foremost, we have stressed the novel contribution of this study. First, the majority of studies in the field are focused around the limited number of model or biotechnologically relevant microorganisms such as E.coli or S.cerevisiae among others, which are prominent for codon usage being an effective measure of translation elongation efficiency of their genes. However, no preferential usage of codons depending on the gene expression level is observed in some organisms such as Helicobacter pylori (Lafay et al., 2000; Sharp et al., 2005), Borrelia burgdorferi (Lafay et al., 1999), and Buchnera aphidicola (Rispe et al., 2004). Therefore, codon usage bias does not work well for a number of organisms (and we do reinforce these arguments in the results of the current study) and beyond the model microorganisms the actual biodiversity in the sense of elongation efficiency optimization types remains largely unexplored. This is part of the reason why we address the phylogeny in Fig. 3 – to preserve the information on evolutionary relatedness between the studied species and to give an idea of what part of this biodiversity of the microorganisms we are talking about. The other reason is that elongation efficiency optimization type is somewhat conservative evolutionary characteristic (we address this part of our investigations in a detail in another yet unpublished paper on case study of bacteria belonging to Ralstonia genus), hence the information about the phylogeny might be useful while dealing with related species to derive some estimates on expected correlation in the absence of protein abundance profile data. Second, it is the first time when the correlation between such elongation efficiency metrics and actual protein abundance has been assessed for a large number of prokaryotes representing various taxonomic groups and we are confident that these results are of interest for the researchers who have a fundamental interest in this field.

It is also worth noting that the main reason why we chose these particular genomes for the further analysis is the availability of genome-wide protein abundance profiles to assess the actual correlation between protein abundance and elongation efficiency indices based on representative datasets, which include protein-encoding genes with various expression levels. Unfortunately, such data is quite scarce because the majority of proteomic studies are focused on a number of particular proteins of interest and their isoforms and their authors do not aim at generating a full protein profile of the organism under study, which we ideally are interested in. We believe that the more studies generating such profiles will be published, the better we will understand the whole pattern and whether uneven coverage in the terms of phylogeny skews it or not. However, we agree that the biological insight of the organisms in terms of EEI matters and hence we have elaborated on that in the revised version of the manuscript.

Unfortunately, we can not agree that the field is nearly saturated with both experimental data and bioinformatic analysis because as we said earlier, these studies predominantly deal with a limited set of model organisms and, speaking of experimental data, protein profiles are unavailable for many organisms. We agree, however, that enlarging the circle of factors that control protein abundance, taking into account the properties of proteins, is important for reaching better understanding in this area. We would like to thank you for mentioning that point. To sum up we have added more focus on the novelty, advancements and key features of this study in the Introduction and Discussion, where it is appropriate, and addressed the points in some other parts of the paper. Now we hope that the resulting manuscript has improved for the general readership.

Additionally, we added the EloE executable, which has been used for the analysis presented in the manuscript, to the Supplementary materials together with ReadMe instructions to make it sustainably available. The article that you have mentioned (Likhoshvai, Yu. G. Matushkin. Nucleotide composition-based prediction of gene expression efficacy. Mol. Biol. (Moscow), 34(3):345–350, 2000) is freely available online: https://link.springer.com/content/pdf/10.1007/BF02759664.pdf . This link is searchable via Google Scholar. EloE was developed initially for the evolutionary studies of microbial genomes. As it was shown in the previous studies, evolutionary optimization of primary structure of protein-encoding genes of unicellular organisms can be classified into the 5 base types depending on the role of codon usage bias and secondary structures in mRNA in determining the translation elongation efficiency. Such a universal approach with the focus on evolution and diversity is original and it is the first paper where the results are presented that validate it using proteomics data specifically. Unfortunately, among the organisms with available protein profiles, Neisseria meningitidis, the only one belonging to the EEI5 base type, do not show a significant correlation between protein abundance and EEI values (not only base ones but any EEI values, including classic codon usage bias), that is why we can not say much about the organisms belonging to that particular optimization type. With more data becoming available, we will be able to shed the light on that question in the future.

Finally, we addressed each of the other minor issues in the revised version of the manuscript. Thank you for the relevant references!

References

Lafay, B., Atherton, J. C., & Sharp, P. M. (2000). Absence of translationally selected synonymous codon usage bias in Helicobacter pylori. Microbiology, 146(4), 851–860. https://doi.org/10.1099/00221287-146-4-851

Lafay, B., Lloyd, A. T., McLean, M. J., Devine, K. M., Sharp, P. M., & Wolfe, K. H. (1999). Proteome composition and codon usage in spirochaetes: Species-specific and DNA strand-specific mutational biases. Nucleic Acids Research, 27(7), 1642–1649. https://doi.org/10.1093/nar/27.7.1642

Rispe, C., Delmotte, F., van Ham, R. C. H. J., & Moya, A. (2004). Mutational and Selective Pressures on Codon and Amino Acid Usage in Buchnera , Endosymbiotic Bacteria of Aphids. Genome Research, 14(1), 44–53. https://doi.org/10.1101/gr.1358104

Sharp, P. M., Bailes, E., Grocock, R. J., Peden, J. F., & Sockett, E. (2005). Variation in the strength of selected codon usage bias among bacteria. Nucleic Acids Research, 33(4), 1141–1153. https://doi.org/10.1093/nar/gki242

Round 2

Reviewer 2 Report

Much improved.